# Reexposure to a sensorimotor perturbation produces opposite effects on explicit and implicit learning processes

**Guy Avraham**[1,2]*, **J. Ryan Morehead**[3,4], **Hyosub E. Kim**[5,6], **Richard B. Ivry**[1,2]

**1** Department of Psychology, University of California, Berkeley, Berkeley, California, United States of America, **2** Helen Wills Neuroscience Institute, University of California, Berkeley, Berkeley, California, United States of America, **3** School of Psychology, University of Leeds, Leeds, United Kingdom, **4** John A. Paulson School of Engineering and Applied Sciences, Harvard University, Cambridge, Massachusetts, United States of America, **5** Department of Physical Therapy, University of Delaware, Newark, Delaware, United States of America, **6** Department of Psychological and Brain Sciences, University of Delaware, Newark, Delaware, United States of America

* guyavraham@berkeley.edu

**Data Availability Statement:** All relevant individual data and summary statistics are within the paper and its Supporting Information. All raw data files

## Abstract

The motor system demonstrates an exquisite ability to adapt to changes in the environment and to quickly reset when these changes prove transient. If similar environmental changes are encountered in the future, learning may be faster, a phenomenon known as savings. In studies of sensorimotor learning, a central component of savings is attributed to the explicit recall of the task structure and appropriate compensatory strategies. Whether implicit adaptation also contributes to savings remains subject to debate. We tackled this question by measuring, in parallel, explicit and implicit adaptive responses in a visuomotor rotation task, employing a protocol that typically elicits savings. While the initial rate of learning was faster in the second exposure to the perturbation, an analysis decomposing the 2 processes showed the benefit to be solely associated with explicit re-aiming. Surprisingly, we found a significant decrease after relearning in aftereffect magnitudes during no-feedback trials, a direct measure of implicit adaptation. In a second experiment, we isolated implicit adaptation using clamped visual feedback, a method known to eliminate the contribution of explicit learning processes. Consistent with the results of the first experiment, participants exhibited a marked reduction in the adaptation function, as well as an attenuated aftereffect when relearning from the clamped feedback. Motivated by these results, we reanalyzed data from prior studies and observed a consistent, yet unappreciated pattern of attenuation of implicit adaptation during relearning. These results indicate that explicit and implicit sensorimotor processes exhibit opposite effects upon relearning: Explicit learning shows savings, while implicit adaptation becomes attenuated

are available from the GitHub repository: https://git.io/Jtiip.

**Funding:** This work was supported by grants NS116883, NS105839 and DC017091 from the National Institutes of Health (https://www.nih.gov/) awarded to RBI. HEK was funded by grants K12 HD055931 from the National Institutes of Health and M3X 1934650 from the National Science Foundation (https://www.nsf.gov/). The funders had no role in study design, data collection and analysis, decision to publish, or preparation of the manuscript.

**Competing interests:** The authors have declared that no competing interests exist.

**Abbreviations:** CI, confidence interval; CR, conditioned response; CS, conditioned stimulus; EEG, electroencephalogram; SEM, standard error of the mean; US, unconditioned stimulus.

## Introduction

Throughout the life span, the motor system needs to learn to correct for errors that emerge due to changes in the state of the body and the environment. When reexperiencing a familiar change, learning can be faster, a phenomenon known as savings upon relearning [1–4]. Constraints on the computations underlying savings in sensorimotor learning have been the subject of considerable debate [5–8]. Recently, converging lines of evidence from visuomotor adaptation tasks indicate that a central component of savings reflects improvement in the use of explicit strategies to counteract an imposed perturbation [9–12]. That is, when participants first encounter a visual perturbation (e.g., rotation of the visual feedback), they may learn to explicitly adjust their behavior to compensate for the perturbation (e.g., aim in the opposite direction of the rotation). Later, upon reexposure to the same perturbation, people quickly recall a successful strategy that had been previously employed, resulting in faster learning.

Yet, the behavioral changes observed during sensorimotor learning do not arise solely from explicit strategy use. The behavioral change in such tasks is also driven by implicit adaptation, the adjustment in the sensorimotor map that occurs outside awareness and volitional control [13]. Indeed, in many contexts, especially those involving small perturbations, most of the learning is implicit [12,14,15]. Whether and how implicit adaptation contributes to savings remains unclear: While some studies have proposed that faster relearning is attributed to implicit processes [6,16,17], others reported no change in the rate of implicit adaptation upon relearning [9,11,12]. Importantly, it can be difficult to obtain a clean assessment of the time course of implicit adaptation across multiple experimental blocks given potential influences from explicit processes and the extended resilience of adaptation over time [18].

Here, we take advantage of protocols specifically designed to isolate explicit and implicit learning processes, asking whether each process is subject to savings upon relearning. We employed a savings design where the same perturbation was imposed during 2 epochs, separated by a washout block that allowed sufficient number of reaches in the absence of the perturbation for the adaptive response to be unlearned [3,8]. In Experiment 1, the visual feedback was rotated by 45˚, and by manipulating the instructions, we obtained separate estimates of explicit aiming and implicit adaptation [12]. In Experiment 2, we used task-irrelevant clamped visual feedback to isolate performance changes resulting from implicit adaptation [19]. The results revealed opposite effects on explicit and implicit motor processes upon relearning: While explicit strategy use improved in response to the second exposure of the perturbation, implicit adaptation was attenuated. A review of the literature and reanalysis of several studies revealed prior, yet unappreciated, evidence that implicit adaptation not only fails to exhibit savings, but also actually becomes attenuated in response to previously experienced errors.

## Results

In Experiment 1, we assessed the contributions of explicit and implicit motor learning processes to savings. Following a baseline block with veridical visual feedback, participants were exposed to the first learning block in which, on Rotation trials, the visual feedback cursor was rotated 45˚ with respect to the position of the hand (Fig 1A). Participants were cued about the perturbed feedback and instructed that their task was to "move the cursor to the target" (Rotation 1). The compensatory response on Rotation trials with this manipulation involves both explicit and implicit processes [12,20]. On a set of randomly selected interleaved trials, the feedback was eliminated, and the participants were instructed to aim directly to the target (Probe 1 trials), with the instructions emphasizing that they should stop using any strategy employed on the Rotation trials. These Probe trials are designed to assay the state of implicit adaptation [12,21,22].

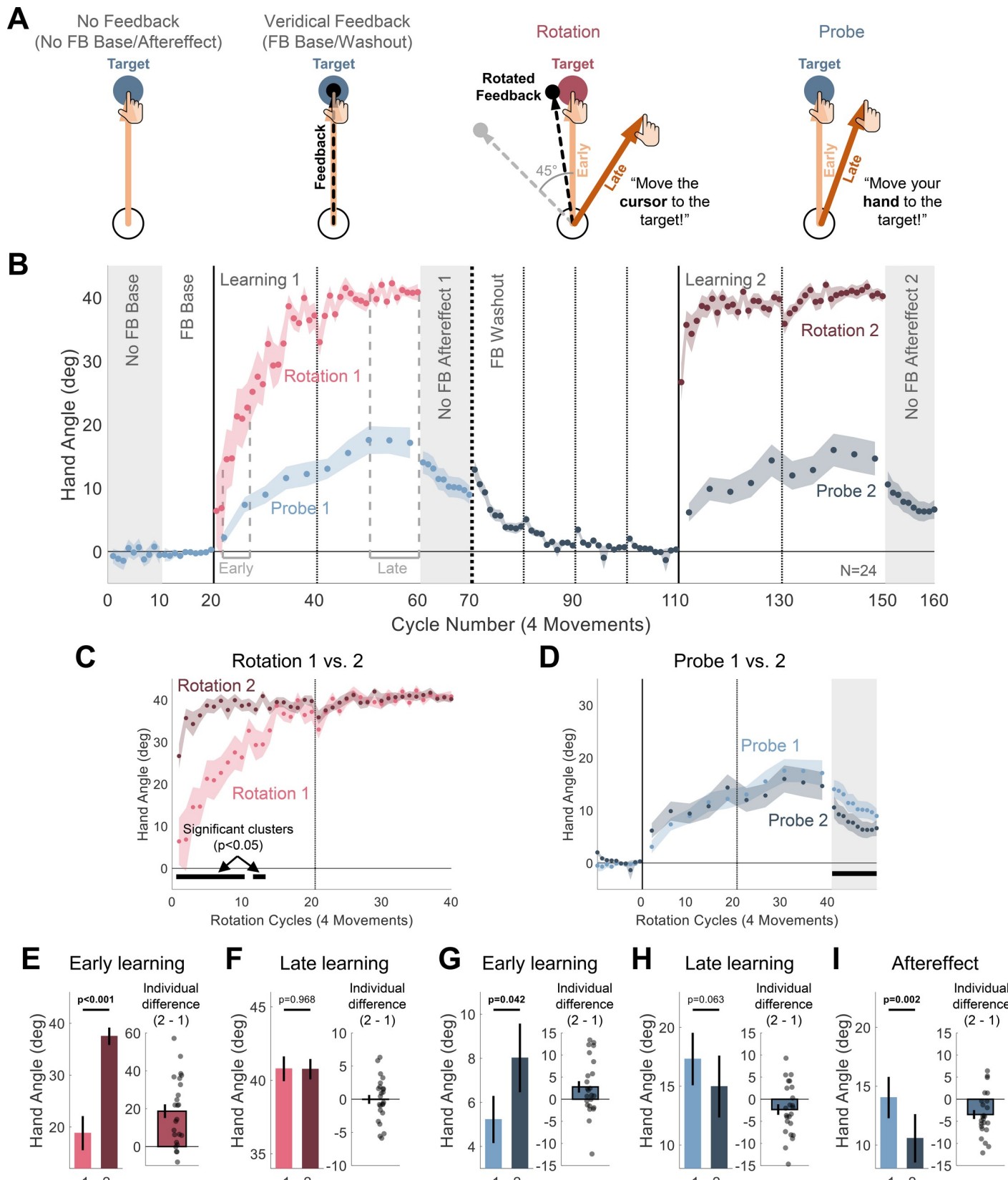

**Fig 1. Experiment 1: Upon relearning a visuomotor rotation, explicit strategies show savings while implicit adaptation is attenuated.** (**A**) Task-level schematics of all trial types. (**B**) Time course of mean hand angle averaged over cycles (4 movements) when participants ($N = 24$) were asked to aim for the target (blue), either during No-Feedback blocks (No FB Baseline and Aftereffect, gray background), Veridical Feedback blocks (FB Baseline and Washout), or No-Feedback Probe trials and when asked to compensate for a rotated cursor (Rotation, pink). Light and dark colors signify blocks 1 and 2 of the experiment. Dotted vertical lines denote 1 (thin) and 2 (thick) min breaks. The labels "Early" and "Late" mark the cycles used for operationalizing measures of early and late learning. (**C, D**) To highlight changes across blocks, overlaid hand angle functions of blocks 1 and 2 for overall learning (explicit and implicit, Rotation trials, C) and implicit adaptation (Baseline, Probe and Aftereffect trials, D). Horizontal thick black lines mark clusters of cycles that show significant difference between the blocks with $p < 0.05$ probability. Cycle numbers in both C and D correspond to the cycles of the Rotation trials. (**E–I**) Summary analysis of Early learning (E, G) and Late learning (F, H) for the Rotation (E, F) and Probe (G, H) conditions, and of the aftereffects (I). Left pair of bars show the mean across participants for each block, and right bar shows the mean of the within-participant differences (Block 2 –Block 1). Black dots indicate individual difference scores. For all figure panels, shaded margins and black vertical lines represent SEM. The individual data and the summary statistics presented in this figure can be found in S1 Data. The raw data can be found in https://git.io/Jtiip. SEM, standard error of the mean.

On Rotation trials, the participants' hands deviated in the direction opposite to the rotation, approaching the ideal 45° change in hand angle by the end of the block ([mean ± standard error of the mean, SEM], 40.8° ± 0.84°, Fig 1B). On Probe trials, the hand angle was also shifted in the same direction, despite instructions to reach directly to the target. By the end of the learning block, this shift was markedly less than that observed on the Rotation trials (17.3° ± 2.23°, t(23) = −8.56, $p < 0.001$, $BF_{10} = 9.91*10^5$, d = −1.75). The large difference in hand angle between the Rotation and Probe trials is consistent with the assumption that the participants employed an explicit aiming strategy to achieve good performance on the Rotation trials and followed the instructions to stop using this strategy on Probe trials (S1 Fig).

Following an extended washout with veridical feedback, the participants experienced a second learning block, again composed of trials with perturbed feedback (Rotation 2), interspersed with no-feedback trials (Probe 2).

Marked savings was observed on the Rotation trials: Performance improved at a faster rate relative to the first learning block (Fig 1C). To statistically evaluate the locus of savings, we performed a series of analyses on the Rotation and Probe trials. For each trial type, 2 approaches were employed. The first, following conventions in the literature, was designed to focus on differences between the 2 learning blocks at predefined stages [3,11]: Early (cycles 3 to 7 of each learning block) and Late (last 10 cycles). Here, we used a 2 (Stage: Early vs. Late) × 2 (Block: Learning 1 vs. Learning 2) repeated measures ANOVA, with planned comparisons targeted at the differences between blocks. In the second analysis, we used a cluster-based permutation test to identify clusters of consecutive cycles that show a significant difference between the 2 learning blocks without relying on predefined assumptions about specific cycles [23,24].

Considering first the data from the Rotation trials, the ANOVA showed main effects of learning within each block (Stage: F(1,23) = 51.4, $p < 0.001$, $\eta_p^2 = 0.53$) and savings across blocks (Block: F(1,23) = 26.2, $p < 0.001$, $\eta_p^2 = 0.69$). The interaction was also significant (Stage × Block interaction: F(1,23) = 25.4, $p < 0.001$, $\eta_p^2 = 0.52$). The increase between blocks, the savings effect, was significant in the early stage ([mean difference ± SEM], 18.7° ± 3.62°, t(23) = 5.16, $p < 0.001$, $BF_{10} = 775$, d = 1.05, Fig 1E), but not in the late stage where performance was near asymptote (0.03° ± 0.67°, t(23) = −0.04, $p = 0.968$, $BF_{10} = 0.215$, d = −0.01, Fig 1F). The cluster analysis revealed that the shift in hand direction was larger in the first quarter of the Rotation trials during the relearning block, relative to first learning block ($p < 0.05$, Fig 1C), consistent with the predefined analysis.

A more complex pattern was observed when analyzing the Probe trials (Fig 1D). In the predefined stage analysis, hand angle increased within each block, indicative of the operation of implicit adaptation (Stage main effect: F(1,23) = 45.0, $p < 0.001$, $\eta_p^2 = 0.66$). However, the effect of block was not significant (F(1,23) = 0.07, $p = 0.800$, $\eta_p^2 = 0.00$). This null result is consistent with the proposition that implicit adaptation does not contribute to savings (see [12]). This conclusion, however, is qualified by a significant interaction (F(1,23) = 8.30, $p = 0.008$,

$\eta_p^2 = 0.27$): The change in hand angle was slightly faster in the early stage of the second learning block (2.79˚ ± 1.30˚, t(23) = 2.15, $p = 0.042$, $BF_{10} = 1.49$, d = 0.44, Fig 1G), but marginally smaller for the late stage (−2.34˚ ± 1.20˚, t(23) = −1.96, $p = 0.063$, $BF_{10} = 1.09$, d = −0.40, Fig 1H). In the cluster analysis, no significant difference was found between the learning blocks.

Although reaction time was not emphasized in this study, these data also point to a dissociation between explicit and implicit processes in savings. Reaction times were smaller on Rotation 2 (695 ± 46 ms) compared to Rotation 1 (858 ± 74 ms) trials (t(23) = −2.93, $p = 0.007$, $BF_{10} = 6.21$, d = −0.60). This decrease is consistent with the idea that strategy use and/or strategy recall improve with experience [25]. In contrast, reaction times did not vary between the first (667 ± 28 ms) and second (667 ± 39 ms) learning blocks for the Probe trials (t(23) = 0.008, $p = 0.993$, $BF_{10} = 0.215$, d = 0.00).

The aftereffect data provide a second measure of implicit learning. Here, the visual feedback was withheld, and the participants were reminded to reach directly to the target. Indeed, these data provide a cleaner measure if we assume participants may occasionally lapse in switching their behavior between the Rotation and Probe trials (see below, [26]). Surprisingly, the aftereffect was weaker after the second block compared to the first in a predefined analysis restricted to the first cycle of each aftereffect block (−3.47˚ ± 1.01˚, t(23) = −3.43, $p = 0.002$, $BF_{10} = 17.2$, d = −0.70, Fig 1I), consistent with the trend observed above for the late stage of the learning blocks. Moreover, the cluster analysis showed a significant reduction of hand angle across the entire second aftereffect block ($p < 0.05$).

To summarize, the results of Experiment 1 show differential effects of relearning on explicit and implicit processes. Clear evidence of savings was observed on the Rotation trials, evident in the early trials where the largest component to learning comes from the use of an explicit aiming strategy [9,12]. In contrast, the overall behavior on the Probe trials was quite comparable between the blocks. There was a modest increase in hand angle early on in the second learning block relative to the first block which might be indicative of savings. However, the design required participants to frequently switch between aiming away from the target (Rotation trials) to aiming to the target (Probe trials). Since the change in aiming is larger during the second learning block, any failure to completely dispense with the aiming strategy would contaminate the Probe trials in a way that would make it appear as if they exhibit savings. Indeed, late in learning, where the behavior on Rotation trials has reached asymptote on both blocks, the difference in hand angle on the Probe trials was actually numerically lower than on the first block.

Importantly, the aftereffect data point to a robust attenuation of implicit adaptation upon reexposure to the perturbation. We considered if this surprising attenuation might be an artifact of the experimental design, and in particular, effects that may occur with the transition between the learning and aftereffect blocks. First, implicit learning is known to involve both labile and stable components, with the former manifest by a reduction in the aftereffect by around 25% over an approximately 60-s delay, even in the absence of any movement [27]. Thus, the attenuation could result from differences in the labile component between the first and second blocks. However, the amount of elapsed time between the end of each learning block and the completion of the first cycle of the aftereffect block was similar for the first and second learning phases. There was a break of approximately 10 s prior to the start of the aftereffect block (during which the experimenter provided the new instructions), and the first cycle of the aftereffect block took about 13 s to complete (Aftereffect 1: 13.6 ± 1.7 s; Aftereffect 2: 12.4 ± 1.7 s; t(23) = −0.710, $p = 0.485$, $BF_{10} = 0.270$, d = −0.15). Thus, the contribution from the labile component should be similar for the 2 aftereffect blocks.

Second, estimates of implicit adaptation can be affected by differential use in aiming [28]. Prior studies have shown that the generalization function of implicit adaptation is centered at

the aiming location. As such, the increase in aiming in the second block could distort the estimate of adaptation in the aftereffect block since the measurements in this block are obtained for reaches to the target location. The mean hand angle in the late phase of the Probe trials are approximately 2° lower on the second learning block. Given that the hand angle is the same on the Rotation trials, this would imply that the aiming location is shifted approximately 2° farther from the target in the second block. Based on estimates of Gaussian generalization functions in the literature (e.g., [29]), a 2° difference in aiming would be expected to produce a miniscule reduction in the aftereffect, approximately 0.5°, or only 14% of the observed reduction in Aftereffect 2. We recognize that this is a crude estimate given that aiming strategy changes over experience, and we do not know the dynamics of aim-based generalization. Despite this limitation, it seems highly unlikely that the reduced aftereffect in the second learning block is due to changes in aiming location.

To obtain a purer assay of the effect of relearning upon implicit adaptation, we conducted a second experiment using a method thought to entirely eliminate changes in aiming and thus avoid issues with switching and generalization. Instead of presenting the feedback at a position contingent on the hand position, Experiment 2 used task-irrelevant clamped feedback [19]. On each reach during the learning blocks (Clamp 1 and Clamp 2), the visual feedback followed a fixed path, rotated 15° (either clockwise or counterclockwise, fixed for a given participant) from the target, with the radial position matched to the participants' hand (Fig 2A). The participant was informed that the angular position of the cursor would be independent of her hand position and that she should ignore the feedback, always reaching directly to the target. Given these instructions, changes in hand angle in response to task-irrelevant clamped feedback is assumed to be implicit, an inference supported by converging evidence from various measures of adaptation [19,30].

The main group of participants (Test group) experienced the clamped feedback over 2 learning blocks, separated by an extended washout block (Fig 2B). As expected, participants showed a robust adaptation effect in both learning blocks, with the heading direction of the hand movement drifting away from the target in the opposite direction of the feedback (Fig 2C). The presumption that this change in behavior was implicit is supported by a number of observations. First, there is a close match between the magnitude of hand angle at the end of the learning block and start of the aftereffect block ([mean Aftereffect/Late learning, 95% confidence interval, CI], Learning 1: [mean, 95% CI], 0.94, [0.88 1.00], Learning 2: 0.85, [0.74 0.96]). The small decay is likely due to some forgetting during the short break between these blocks [18,27,31]. Second, consistent with the hypothesis that participants aimed directly to the target throughout the experiment, reaction times were uniformly fast (approximately 400 ms) and comparable across different phases of the experiment (Clamp vs. Aftereffect: $t(15) = 0.810$, $p = 0.430$, $BF_{10} = 0.340$, d = 0.20; Learning 1 vs. Learning 2: $t(15) = -0.390$, $p = 0.702$, $BF_{10} = 0.273$, d = -0.10). Third, in a debriefing survey, all of the participants reported that they always planned their reaches to the target, and none reported awareness of the systematic changes in hand direction.

We used the same 2-part statistical analysis as in Experiment 1, first analyzing the data from prespecified blocks and then with the cluster analysis that considers all of the data. The ANOVA showed a main effect of stage ($F(1,15) = 53.4$, $p < 0.001$, $\eta_p^2 = 0.78$), with hand angle increasing between the early and late phases of a block. Importantly, the effect of block was also significant ($F(1,15) = 10.8$, $p = 0.005$, $\eta_p^2 = 0.42$), with the adaptation function markedly attenuated in the second learning block (Fig 2D). Although the interaction was not significant ($F(1,15) = 0.76$, $p = 0.399$, $\eta_p^2 = 0.05$), we conducted our planned comparisons for each learning stage. In the early phase, adaptation was numerically lower, but the difference was not significant ([mean difference ± SEM], $-1.82° ± 1.33°$, $t(15) = -1.37$, $p = 0.191$, $BF_{10} = 0.562$, d =

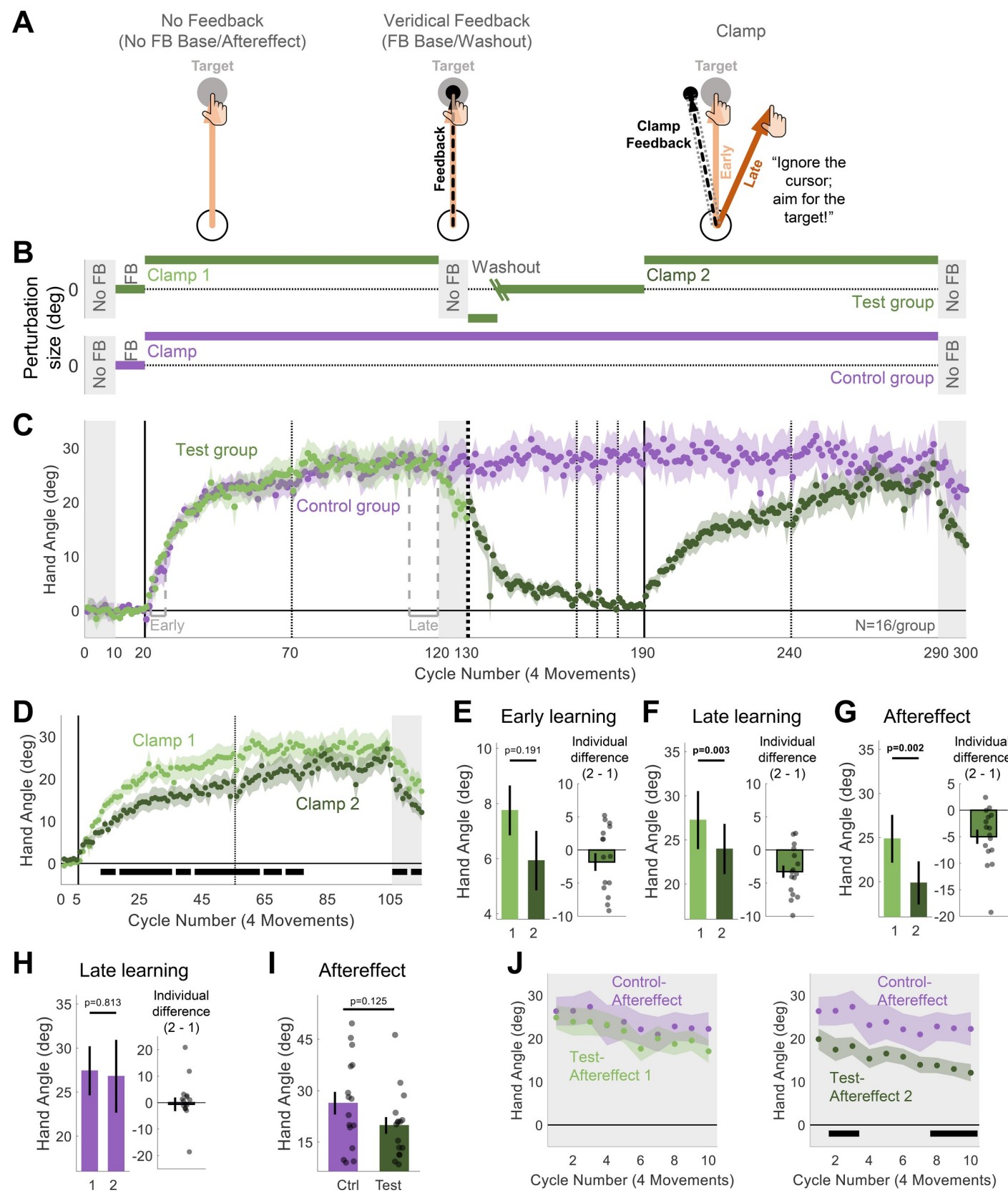

**Fig 2. Experiment 2: Task-irrelevant clamped feedback revealed an overall attenuation of implicit adaptation upon relearning.** (**A**) Task-level schematics of all trial types. (**B**) Experimental protocol of 2 experimental groups: Test (*N* = 16, green) and Control (*N* = 16, purple). For the Test group, the green oblique lines in the Washout block represent a transition from a reversed-clamp phase to a veridical feedback phase; the cycle of the transition was determined based on each individual's performance in the reversed-clamp phase (see Methods). (**C**) Time courses of mean hand angle averaged over cycles (4 movements) for both groups. For the Test group, light and dark green colors signify blocks 1 and 2 of the experiment, with the onset of the task-irrelevant clamped feedback marked by the vertical solid lines. Dotted vertical lines denote 1 (thin) and 2 (thick) min breaks. The labels "Early" and "Late" mark the cycles used for operationalizing measures of early and late learning. (**D**) Overlaid hand angle functions for the 2 blocks. Horizontal thick black lines denote clusters that show a significant difference between blocks 1 and 2 (*p* < 0.05). (**E–H**) Summary analysis of Early learning (E), Late learning (F), and Aftereffect (G) stages for the Test group and Late learning for the Control group (H). Left pair of bars show the mean across participants for each block, and right bar shows the mean of the within-participant differences (Block 2 –Block 1). Black dots indicate individual difference scores. (**I**) Summary analysis comparing the Aftereffect between the groups in the last No FB block. Black dots represent data of individual participants. (**J**) Overlaid hand angle functions for the Aftereffect block of the Control group and the Aftereffect 1 (left panel) or Aftereffect 2 (right panel) of the Test group. For all figure panels, shaded margins and black vertical lines represent SEM. The individual data and the summary statistics presented in this figure can be found in S1 Data. The raw data can be found in https://git.io/Jtiip. SEM, standard error of the mean.

−0.34, Fig 2E). In the late phase, the attenuation of adaptation was significant (−3.28° ± 0.91°, t (15) = −3.59, *p* = 0.003, $BF_{10}$ = 16.3, d = −0.90, Fig 2F). A significant attenuation effect was also found in the planned comparison of the aftereffect data (−5.00° ± 1.34°, t(15) = −3.74, *p* = 0.002, $BF_{10}$ = 21.5, d = −0.94, Fig 2G). The cluster analysis provided a similar picture. Implicit adaptation was attenuated throughout much of the learning and aftereffect blocks (*p* < 0.05, Fig 2D).

Given that attenuation of implicit adaption in relearning has not been highlighted in the literature, we conducted a replication study using an online platform [32,33], one in which the crowdsourced participants complete the study remotely using their individual display and response devices. The results from the online study were quite similar to those observed in Experiment 2 (S2 Fig, all statistics are reported in the caption). Of particular note, adaptation was lower during the second block in the late phase of learning ([mean difference ± SEM], −1.57° ± 0.74°, t(47) = −2.14, *p* = 0.038, $BF_{10}$ = 1.25, d = −0.31) and in the aftereffect (−3.35° ± 0.81°, t(47) = −4.14, *p* < 0.001, $BF_{10}$ = 168, d = −0.60), and the attenuation was observed in multiple clusters throughout the learning and aftereffect blocks (*p* < 0.05). Considered in tandem with the results of Experiment 2, the results clearly show that implicit adaptation elicited in response to clamped feedback is attenuated upon relearning.

We considered the possibility that the attenuation might reflect fatigue or some sort of a habituation process, one in which the response to the clamped error signal becomes attenuated with extended use. To test this hypothesis, a separate group of participants (Control group) was exposed to a single, extended block of clamped feedback with no washout (Fig 2B). The participants from this group showed near-identical adaptation to that observed in the Test group over the cycles corresponding to the first learning block (Fig 2C). Moreover, they remained at this asymptote for the remainder of the experiment, with no indication of a shift in hand angle back toward the target. This was confirmed in an analysis comparing the hand angle for the Control group at cycles corresponding to the late stages for the first and second learning blocks in the Test group (cycles 111 to 120 vs. cycles 281 to 290, −0.62° ± 2.58°, t(15) = −0.24, *p* = 0.813, $BF_{10}$ = 0.262, d = −0.06, Fig 2H).

Finally, we performed a direct comparison of the aftereffect data from the Test and Control groups. In the predefined analysis (cycle 291 only), the magnitude of the aftereffect was approximately 25% lower in the Test group (19.9° ± 2.45°) than in the Control group (26.4° ± 3.31°), although this difference was not significant (t(30) = −1.58, *p* = 0.125, $BF_{10}$ = 0.587, d = −0.56, Fig 2I). For the cluster analysis, we did 2 analyses, comparing the Control group data from the single aftereffect block (cycles 291 to 300) to the Test group data from the first aftereffect block (cycles 121 to 130) and the second aftereffect block (cycles 291 to 300). There were no significant clusters when the Test group data came from the first aftereffect block (Fig 2J, left panel). In contrast, multiple time points in the cluster analysis showed an attenuation effect

when the Test group data came from the second aftereffect block (Fig 2J, right panel). These results indicate that the attenuation of implicit adaptation takes place specifically upon relearning and is not attributable to fatigue or habituation.

## Discussion

Savings is a relatively ubiquitous phenomenon, observed in domains as diverse as classical conditioning [34], procedural learning [35], and associative memory [36]. Within the domain of sensorimotor learning, savings has been observed in adaptation studies involving perturbations of arm movements [3,37], locomotion [38], and saccadic eye movements [39]. Recent work has taken a more mechanistic approach, seeking to specify constraints on savings. One key insight here is that the benefits observed during relearning may be limited to certain component processes of learning; in particular, savings has been associated with explicit strategy use, but not implicit sensorimotor adaptation [9,12,40]. Our results go beyond this observation, indicating that implicit adaption not only fails to exhibit savings, but also is actually attenuated upon relearning. Taken together, relearning a visuomotor transformation appears to produce opposite effects on explicit strategies and implicit adaptation: While the explicit system exhibits savings, the implicit system shows attenuation.

### Attenuation of implicit adaptation upon relearning is a robust phenomenon

The finding that implicit adaptation becomes attenuated upon relearning stands in contrast to prior reports that have either reported an absence of savings (i.e., no change upon relearning, e.g., [12]) or signatures of savings (e.g., [17]). An absence of savings has been interpreted as reflective of the inflexible nature of the implicit adaptation system [14,19,41,42], consistent with other findings showing that adaptation is minimally influenced by top-down factors such as task outcome [43]. By contrast, evidence showing savings during adaptation motivated a model in which the system retains a memory of previously experienced errors and exhibits increased sensitivity to those errors when reencountered [6]. This model can explain behavioral changes in the rate of learning to errors of different size [44], as well as manipulations of the learning context [6,45]. However, these behavioral effects may not reflect the memory of errors per se, but rather, the contribution of explicit learning processes that, through the recognition of a particular learning context, recall context-specific strategies that compensate for an associated perturbation [12,14,41].

Prior to initiating this project, we were not aware of any reports of attenuation in implicit adaptation during relearning. In retrospect, this may reflect a bias in the literature to look for savings given the ubiquity of this phenomenon in the learning literature. Moreover, as noted in our motivation for Experiment 2, it can be difficult to obtain "pure" measures of implicit learning. Learning in standard adaptation tasks tends to reflect the composite operation of multiple processes, and it is just in the past decade that analytic methods have been developed to cleanly partition the learning benefits into its component sources [9,20,46–48].

Given the unexpected, yet consistent, pattern of attenuation observed in the experiments reported here, we conducted a review of the literature, seeking to determine if this phenomenon had been overlooked in previous studies. Our primary inclusion criteria were (1) that the design included 2 learning phases with some form of washout between the 2 phases; and (2) the learning data of interest were estimates of implicit adaption with limited explicit contribution. We note that in most of these studies, the analysis relevant to our review was either not reported or was only noted in passing since the focus was on a different question.

We identified 13 experiments from 9 papers that met these inclusion criteria. For each experiment, we calculated the change in performance in relearning, using Cohen's *d* as a standardized metric of effect size to facilitate comparison across experiments. In some studies, it was possible to make (or obtain) this calculation from the published work; in the other instances, we obtained the data from the authors, performed the requisite analyses, and then calculated Cohen's *d*. Consistent with the current results, the overall pattern in these experiments was one of attenuation of implicit adaptation upon relearning, and in the majority of the cases, the effect size was moderate or higher (Fig 3A). The attenuation effect was most

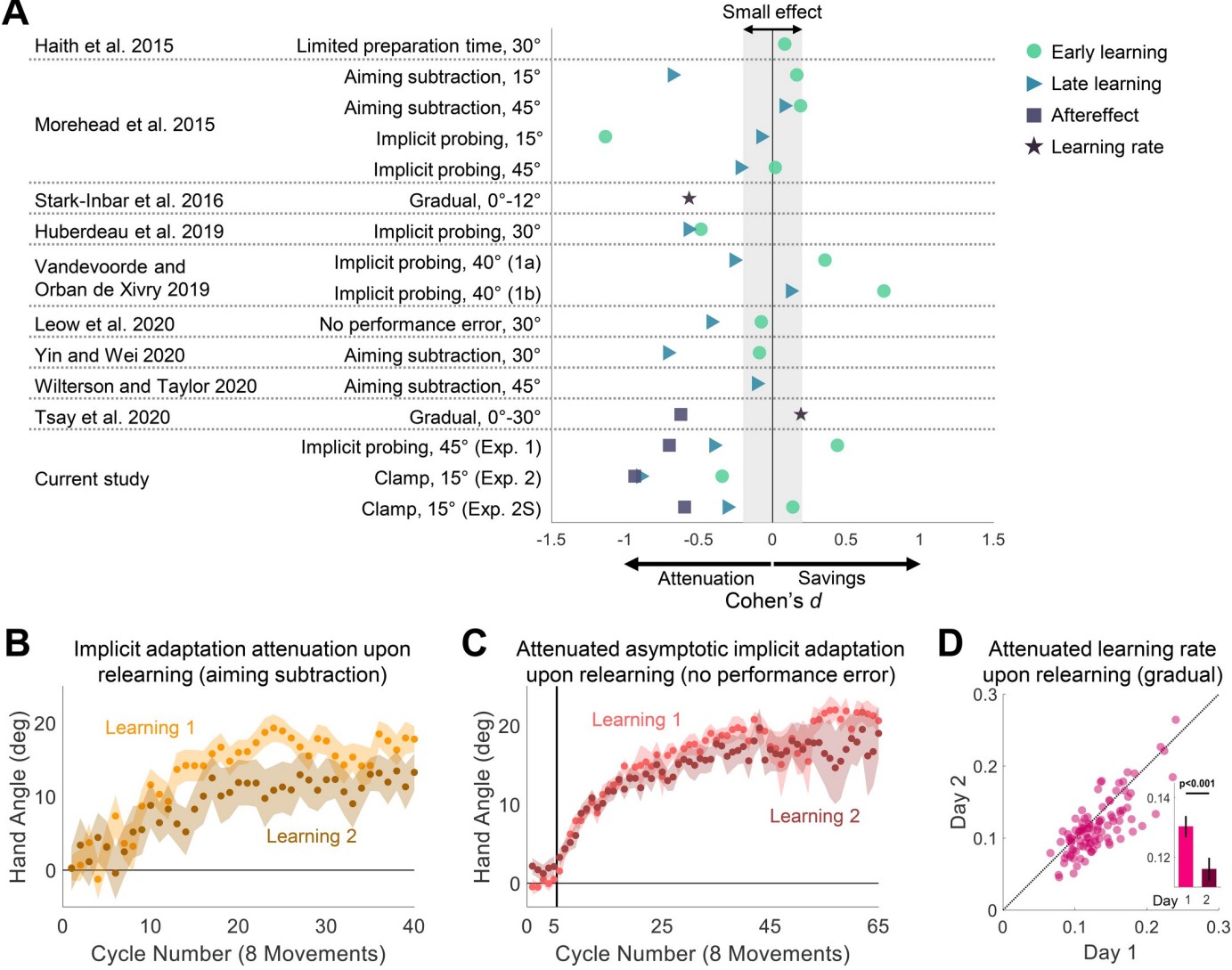

**Fig 3. Prior evidence for attenuation upon relearning for implicit visuomotor adaptation.** (A) Cohen's *d* effect size of the differences in learning measures between the second and first learning blocks for experiments meeting our criterion for inclusion in the review of the literature. Early learning (green circles), Late learning (blue triangles), Aftereffect (purple squares), and Learning rate (dark purple stars). (B, C) Overlaid hand angle functions of implicit adaptation to a visuomotor rotation over 2 learning blocks. Light and dark colors denote blocks 1 and 2 of the experiment, respectively. Shaded margins represent the SEM. In **B**, implicit adaptation was extracted by subtracting a reported aiming location from movement hand angle on every trial. Adapted with permission from Yin and Wei (2020). In **C**, implicit adaptive response to the rotated cursor when the target jumps in a manner that eliminates task error. Adapted with permission from Leow et al. (2020). (**D**) Learning rate during adaptation to a gradually changing visuomotor rotation, tested on 2 days. Pink markers represent individual participants. Black diagonal dotted line represents the unity line. Bars and black vertical lines (inset) represent mean and SEM, respectively. Adapted with permission from Stark-Inbar et al. (2016). The individual data and the summary statistics presented in this figure can be found in S1 Data. SEM, standard error of the mean.

pronounced when the experiments included a test for late learning or had an aftereffect block; the results for early learning were inconsistent. We suspect the inconsistency in the early learning results arises in part, from the fact that the level of learning is generally low during this phase, making it less sensitive for revealing changes between blocks. Moreover, any failure to completely wash out the effects of adaptation from the first block will introduce a bias toward savings.

To detail a few examples, consider first a recent study by Yin and Wei [17]. The focus of the paper was on a generalization issue, showing that following exposure to conditions conducive to implicit adaptation in an initial learning block (e.g., clamped feedback), participants learned at a faster rate when presented with an abrupt 30˚ visuomotor rotation (relative to a group who did not receive the initial training). While the behavior here could be seen as a form of savings, it is not clear that the savings came from a change in the implicit process. First, the introduction of the 30˚ rotation in the second block engages both explicit and implicit processes. Second, veridical feedback was used to washout learning at the end of the first block. This introduces a salient error at the start of washout which might impact subsequent behavior in the second learning block. More relevant to the present discussion is the performance of their control group ([17], Experiment 2). For this group, the same 30˚ rotation was presented in 2 blocks separated by a washout period (i.e., a classic savings design). Importantly, participants reported their aiming location prior to each reach, and thus, implicit adaptation could be inferred by subtracting the aiming location from the actual hand position [20]. As shown in Fig 3B, the estimate of the implicit component was attenuated in the second learning block. A similar pattern was observed in 2 other studies using the aiming report task [12,49].

A second example comes from a paper by Leow and colleagues [11]. Here, the authors used a task in which the target was displaced during the reach such that it intersected the perturbed cursor. This elimination of task error (the cursor "hits" the target) has been argued to provide another method for isolating implicit learning [47]. When this task was repeated over 2 blocks, separated by a washout period, the asymptotic level of adaptation was attenuated in the second block (Fig 3C).

The third example comes from our own lab, an individual difference study in which participants were tested on a variety of implicit learning tasks, each repeated over 2 sessions [50]. The battery included a visuomotor rotation task that used a schedule in which the perturbation changed in a gradual manner to minimize awareness. The estimated mean learning rate of the participants was lower on the second day (Fig 3D). Attenuation was also observed in another study from the lab using a gradual perturbation schedule [51].

Our reanalysis of the literature also identified experiments that either showed no significant change in implicit adaptation upon relearning or even savings (see Fig 3A). Notably, all of the cases showing a moderate effect size in the direction of savings used the design of Experiment 1, where participants alternate between Rotation and Probe trials ([12], Experiment 4; [22], Experiments 1a and 1b). As noted previously, this method may be problematic for isolating implicit learning due to contamination on Probe trials if the participant fails to dispense with the aiming strategy on some trials. This issue would be most salient in the early stages of learning when differences between strategy use are especially large between the 2 learning blocks. Indeed, the "savings" effect was mainly found during early learning in these experiments.

We do not take the current results to be indicative of constraints on relearning for all forms of implicit learning. Savings-like phenomena have been observed on implicit tasks involving de novo learning, as in mirror drawing and eyeblink conditioning, or when recalibrating an existing representation, as in saccade adaptation. In one of the most widely cited examples, the learning function on a mirror drawing task for the amnesia patient H.M. resumed each day around the same level as where it had ended on the previous session, despite his lack of

awareness of having done the task before [52]. While there was no "washout" phase to provide a real test of savings, there was no indication that the rate of learning weakened over sessions. Savings has also been well documented in the eyeblink conditioning literature [53–57] and in saccade adaptation [39].

## Possible mechanisms for attenuation upon relearning

We consider 2 general explanations for an attenuation of relearning rate. First, we discuss how attenuation could come about from the modulation of parameters governing the alteration of a single memory by the error experienced on each movement. We then turn to a hypothesis by which attenuation reflects the interplay of multiple memories. Given that implicit sensorimotor adaptation is paradigmatic as a cerebellar-dependent process [19,58–64], we make reference to other cerebellar-associated learning phenomena in our discussion of these mechanistic hypotheses.

Within the context of sensorimotor adaptation, savings has traditionally been modeled in terms of changes that impact a unitary memory of a sensorimotor map. The classic example is the single-rate state-space model [65,66]. Here, trial-by-trial learning reflects the operation of 2 parameters, a retention factor corresponding to how well the system retains a memory of its current state and a learning rate corresponding to how that state should be modified given the error just experienced. With this model, savings can be explained by an increase in retention [67] and/or an increase in the learning rate [6,8] across learning blocks. Similarly, eyeblink conditioning is typically modeled with the Rescorla–Wagner model, which formalizes the trial-by-trial change in the associative strength between the conditioned and unconditioned stimuli in terms of a learning rate parameter and the salience of the conditioned stimulus (CS). Thus, savings could arise from the faster operation of the associative process, or could be attentional in nature, with the salience of the CS amplified when reencountered in a familiar context [68,69].

The same models that adjust parameters of a state update equation over experience can account for attenuation in relearning; in the state-space model, attenuation would result if the retention factor or learning rate were reduced. Indeed, a number of factors that attenuate learning have been modeled this way, including the effect of visual feedback uncertainty [70–72] and task success [47,73,74].

As shown in the current study, implicit adaptation may be attenuated merely by reexposure to a perturbation. A simple experience-dependent reduction in either the learning rate or retention parameters is ruled out because the Control group in Experiment 2 showed no attenuation in the magnitude of their asymptotic learning. However, attenuated adaptation upon relearning, as observed in the Test group, could reflect a desensitization process, one in which the system becomes less sensitive to a familiar error when, after washout, that error is reencountered. Mechanistically, desensitization in readaptation of the vestibulo–ocular reflex has been attributed to the saturation of recently activated synapses [75]. Applying this idea to our results, we might suppose that mechanisms underlying adaptation become saturated during the initial learning block. As long as the environment does not change (as in the Control group), the system can remain at asymptote. However, desensitization would occur during relearning if the washout phase did not allow sufficient time to rejuvenate synaptic elements required for plasticity. This hypothesis predicts that the degree of attenuation would be dependent on the duration of the washout phase: Less attenuation should be seen if the duration of the washout phase is increased. However, the fact that attenuation of implicit adaptation is observed across days [49–51] seems problematic for this unitary memory hypothesis.

An alternative framework for understanding attenuation during relearning is to consider how learning functions may be modulated by the interplay of multiple memories. Various

reformulations of the state-space model have been developed to incorporate this idea. For example, the basic state-space model can be expanded to allow for multiple learning processes that operate on different timescales ([7]; see also [76]). Interestingly, the interplay of fast and slow processes, even when their associated parameters remain fixed, can produce savings, at least when the washout period is not sufficiently long to allow both processes to return to the null state [8].

In a complementary way, attenuation in relearning may also reflect the interplay of component processes that have opposing effects on behavior. In studies of sensorimotor adaptation, interference is observed when opposing perturbations alternate in a random manner [77], as well as when presented sequentially [78,79]. In such situations, the absence of savings has been attributed to interference between 2 internal models, one associated with each perturbation [3]. Similarly, savings is abolished in saccade adaptation with a protracted washout period [39], as well as in conditioning studies when the extinction phase involves the repeated presentation of the CS and the unconditioned stimulus (US) in an unpaired manner [80,81]. Theoretically, in these manipulations, a reversed perturbation or decoupling of the CS–US relationship leads to the establishment of a new memory, one that produces interference when the original situation is reencountered [82,83].

Interference can, of course, not only abolish savings, but also produce attenuation in relearning when the irrelevant memory trace continues to contribute to performance. This idea has been extensively examined in the classical conditioning literature. For example, in fear conditioning, attenuation is observed when there is an extended extinction period involving just the presentation of the CS alone (e.g., tone that had been associated with a foot shock) [84]. The reduced rate of learning observed when the CS is again paired with the US has been modeled as resulting from competition between 2 memories, one reflecting the paired CS–US association leading to a conditioned response (CR) and the other the solitary CS, associated with no response. This model nicely accounts for the fact that attenuation is especially pronounced when the extinction period is long and when the context remains unchanged [85,86].

It remains to be seen if similar constraints are relevant to the attenuation observed in the current study. Given concerns with residual effects from the first adaptation block, we designed the studies to ensure a strong washout phase: We used a large number of veridical feedback trials in Experiment 1 and a reversed clamp, followed by veridical feedback in Experiments 2 and 2S. These manipulations would be conducive sources of interference (e.g., extended association with veridical feedback in the experimental context). We note that the multi-session study of Wilterson and Taylor [49] did not include a washout phase, and the interference hypothesis would not seem a viable explanation for the modest attenuation observed across sessions (see Fig 3A). However, interference here may be between the laboratory and natural motor behavior that occurs once the participant has left the lab between sessions.

To this point, we have elaborated on 2 general models of how learning might be modulated as a function of experience: changes in the parameters of a single memory or an interplay between different learning processes, each having a specific set of stable parameters. A hybrid of these 2 approaches has been offered to explain savings in situations where neither approach is sufficient on its own [4,8]. For example, in an extension of the multi-rate idea, Hadjiosif and Smith [18] proposed that savings was restricted to the fast, labile process, with the rate of learning for this process increased upon reexposure to a perturbation.

The distinction we observed between explicit and implicit learning processes in terms of savings can also be understood from this hybrid perspective. It seems clear that explicit processes such as aiming can, and do, exhibit savings [9,11,12]. While this could be modeled by postulating a faster learning rate or stronger retention rate, the underlying psychological

process is likely one of memory retrieval. When reexposed to a perturbation in a familiar context, the participant recalls a successful strategy [12]. Using methods that allowed us to isolate the contribution of implicit processes revealed the opposite behavioral profile here, one of attenuation during relearning. Determining if this attenuation arises from a desensitization of an implicit process (or processes) or interference from the implicit interplay of multiple memories remains a question for future study.

## Methods

### Ethics statement

The study was approved by the Institutional Review Board at the University of California, Berkeley (Protocol 2016-02-8439) and adhered to the principles expressed in the Declaration of Helsinki. All participants provided written informed consent to participate in the study.

### Participants

A total of 104 healthy volunteers (aged 18 to 40 years; 64 females) were tested: 24 in Experiment 1, 32 in Experiment 2, and 48 in Experiment 2S. In Experiments 1 and 2, all participants were right-handed, as verified with the Edinburgh Handedness Inventory [87]. The participants in Experiment 2S did not complete a handedness inventory, but 42 of them self-reported being right-handed, 4 left-handed, and 2 ambidextrous. The sample size for each experiment was set to include approximately 30% more participants than the minimum required to ensure good statistical power and appropriate counterbalancing of the perturbation direction and target location sets (details provided in the protocol for each experiment).

### Experimental setup and task

The participant sat at a custom-made table that housed a horizontally mounted LCD screen (53.2 cm by 30 cm, ASUS), positioned 27 cm above a digitizing tablet (49.3 cm by 32.7 cm, Intuos 4XL; Wacom, Vancouver, Washington, United States of America). Stimuli were displayed on the LCD screen. The experimental software was custom written in MATLAB (The MathWorks, Natick, Massachusetts, USA) using the Psychtoolbox extensions [88].

The participant performed center-out movements by sliding a modified air hockey paddle containing an embedded digitizing stylus across the tablet. The tablet recorded the position of the stylus at 200 Hz. The monitor occluded direct vision of the hand, and the room lights were extinguished to minimize peripheral vision of the arm.

At the beginning of each trial, a white circle (0.5-cm diameter) appeared on the center of the screen, indicating the start location. The participant moved her hand (holding the stylus) to the start location. Feedback of hand position was indicated by a white cursor (0.3-cm diameter) and only provided when the hand was within 1 cm of the start location. There was an approximately 33-ms delay between the sampling of the tablet and update of the cursor position based on that sample. After the hand position was maintained in the start location for 500 ms, a colored target (0.5-cm diameter circle) appeared at one of 4 locations around a virtual circle, with a radial distance of 8 cm from the start location. Within each experimental group, the target locations were in a diagonal layout, at 45˚ (0˚ being the positive edge of the abscissa), 135˚, 225˚, and 335˚ for half of the participants and in an oblique layout, at 20˚, 110˚, 200˚, and 290˚ for the other half of the participants.

The participant was instructed to rapidly move her hand following the presentation of the target, attempting to slice through the target with either the hand or cursor (depending on the specific experimental protocol). Movement time was calculated as the interval between the

time at which the amplitude of the movement exceeded 1 cm from the start location to the time at which the amplitude reached a radial distance of 8 cm, the target distance. To encourage the participants to move fast, the auditory message "too slow" was played if the movement time exceeded 300 ms. Participants had little difficulty meeting this criterion, with an overall mean movement time across the 2 experiments of 132 ± 4.16 ms (± SEM). After moving to the target, the participant moved back to the start location. The overall mean time required to return to the start position was 1.69 ± 0.05 s.

## Experimental protocol

The primary goal of the study was to evaluate changes in implicit adaptation upon relearning. Thus, the general design in each experiment incorporated 2 learning blocks that were separated by a long washout block.

**Experiment 1.** A total of 24 participants were tested in Experiment 1. The target number of participants was determined based on the effect size (d = 0.67) observed for savings in a previous study that used a similar protocol [12], with a significance level of $\alpha$ = 0.05 and power of 0.8. This analysis yielded a minimum sample size of 16 participants. The tested number was increased given a criterion to increase the sample size by at least 30% above the minimum and counterbalancing considerations.

The experimental session consisted of 180 movement cycles, each involving 4 movements, one to each target. As mentioned above, the targets were positioned either in a diagonal or oblique layout. A diagonal layout is commonly used in visuomotor rotation studies [20,89]. However, we were concerned that a 4-target diagonal layout might be conducive for finding an optimal aiming solution to counteract a 45˚ rotation (see below) since these would require movements along the cardinal directions (in the absence of implicit adaptation). For this reason, we opted to use an oblique layout for half of the participants.

The session was divided into the following blocks (Fig 1B): No-Feedback Baseline (10 cycles), Veridical Feedback Baseline (10 cycles), Learning 1 (50 cycles), Aftereffect 1 (10 cycles), Washout (40 cycles), Learning 2 (50 cycles), and Aftereffect 2 (10 cycles). The initial baseline blocks were included to familiarize the participants with the experimental setup. In these trials, participants were instructed to move their hand directly to a blue target. Veridical feedback was provided in the second baseline block to minimize idiosyncratic reaching biases.

Two types of trials were randomly interleaved during the learning blocks: Rotation and Probe trials. For the Rotation trials (40 cycles per block), the position of the cursor was rotated by 45˚ with respect to the position of the hand. We chose a rotation size of 45˚ because adaptation to a perturbation of this size reliably produces savings upon relearning [8,12]. The direction of the rotation (clockwise or counterclockwise) was fixed for a given participant and counterbalanced across participants. On Rotation trials, the color of the target was red, providing a cue that the cursor position would be perturbed. At the start of the block, the participant was instructed that a red target signified that the cursor "will act differently" and that their goal was to make the cursor hit the target.

For the Probe trials (10 cycles per block), the color of the target was blue. The participant was instructed that the cursor would not be visible on these trials and that their goal was different: Now they were to reach directly to the blue target, discontinuing any strategy they might have adopted when reaching to the red targets. To emphasize these instructions, the message "Move your hand to the target" appeared simultaneously with the blue target. The position of the instruction was either above or below the center of the screen, selected on each trial to be closest to the target (above for targets between 0˚ and 180˚ and below for targets between 180˚ and 360˚).

To minimize possible effects related to switching between the 2 types of trials, the participants were informed that, while they could initiate the reach at any time after the onset of the target, they should take their time to comply with the instructions/goal associated with each target color. After cycles 40 and 130, the midpoints of the learning blocks, the participant was provided with a 1-minute break.

To provide another measure of implicit adaptation, following each of the learning blocks, we included aftereffect blocks (1 and 2) in which the feedback was eliminated in all trials. Just before the start of the aftereffect block, the participant was informed that the cursor would no longer be visible and that the task was to move her hand directly to the target. There was no additional break prior to the start of the aftereffect block, minimizing the time for learning to decay.

The washout block was introduced following the first aftereffect block and a 2-minute rest period. During this block, the participant was instructed to reach directly to the target, and veridical feedback was provided on each trial with the aim of bringing the sensorimotor map back to a baseline state. The minimal number of washout cycles (40) was set to be the same number as that used during the learning block and should be sufficient to ensure full unlearning [12]. We included three 1-minute breaks during the block to verify that the effects of implicit adaptation were completely washed out by the beginning of the second learning block. These probes have been shown to be useful in revealing residual effects of adaptation, manifested as transient increases in hand angle after the break [22].

**Experiment 2.**  Participants in Experiment 2 were assigned to either the Test ($N = 16$) or Control ($N = 16$) condition. The target number of participants was determined based on the effect size (d = 0.70) of the attenuation effect observed in the aftereffect data in Experiment 1. This analysis yielded a minimum sample size of 12 participants, with the actual number tested based on the criterion to increase the sample size by at least 30% above the minimum and counterbalancing considerations.

The experimental session consisted of 300 movement cycles of 4 movements (to 4 target locations, see above). For the Test group, the session was divided into the following blocks (Fig 2B): No-Feedback Baseline (10 cycles), Veridical Feedback Baseline (10 cycles), Clamp 1 (100 cycles), Aftereffect 1 (10 cycles), Washout (60 cycles), Clamp 2 (100 cycles), and Aftereffect 2 (10 cycles).

To isolate implicit adaptation, we used task-irrelevant clamped feedback in the Clamp blocks. Here, the cursor moved in a fixed path, displaced 15˚ from the target (clockwise or counterclockwise, fixed for each participant and counterbalanced). The radial position of the feedback cursor corresponded to the participant's hand, but the angular position of the cursor was invariant, independent of the participant's reaching direction. The participant was instructed to ignore the feedback and always reach directly to the target. To make this salient, 2 demonstration trials were performed prior to each clamp block. On each demonstration trial, the target appeared at the 78˚ location. For the first trial, the participant was told to "Reach straight to the left" (180˚); for the second trial, the participant was told to "Reach backwards towards your body" (270˚). On both trials, the cursor trajectory was clamped, following the same 15˚ path off from the target that the participant would experience throughout the Clamp blocks. The clamped feedback offset of 15˚ falls within the range of error sizes that induce an invariant rate and magnitude of adaptation (approximately 6˚ to 60˚ [42]).

Aftereffect blocks with no feedback were introduced immediately following each of the Clamp blocks. The participant was informed before each block that the cursor would not be visible and was instructed again to move directly to the target.

Rather than using veridical feedback in the washout block (as in Experiment 1), we adopted a different procedure to eliminate the effects of adaptation in Experiment 2. The introduction

of veridical feedback after the conclusion of the first clamp block would result in a relatively large discrepancy between the expected and observed feedback, assuming adaption has occurred. We were concerned that this would make participants aware of the change in behavior and that this might alter their response to the clamped feedback in the second clamp block (e.g., invoke a strategy to offset the anticipated effects of adaptation). To minimize awareness of adaptation, the washout block consisted of 2 phases. In the first phase, we reversed the direction of the clamp. The participant was informed that she would again not have control over the movement direction of the feedback and reminded to ignore the feedback, aiming directly for the target. This manipulation induced a reversal in adaptation and thus drove the direction of the hand back toward the target. When the median reach direction was within 5° of the target for 6 consecutive cycles, the second phase was implemented. Here, the feedback became veridical. Given that the total number of washout cycles was fixed, the number of cycles in each phase was determined on an individual basis using the performance criterion described above. All of the participants in the Test group experienced at least 30 cycles (37.3 ± 1.62) of veridical feedback before the second clamp block, ensuring they had sufficient exposure to unperturbed feedback prior to the onset of the second learning block. Demonstration trials were provided at the start of each phase of the washout block, 2 for the reversed clamp and 2 for veridical feedback. The demonstration trials were similar to those presented before the clamp blocks (same target location and same instructions for where to reach). The provided feedback was matched to the feedback in the subsequent phase (reversed clamp/veridical). Note that the demonstration trials for the veridical feedback phase appeared in the transition between the phases, when the participant already reached close to the targets.

The Control group was included to provide a between-group comparison to the performance of the Test group during the second clamp block. For the Control group, the session was divided into the following blocks (Fig 2B): No-Feedback Baseline (10 cycles), Veridical Feedback Baseline (10 cycles), extended Clamp (270 cycles), and No-Feedback Aftereffect (10 cycles). The 2 demonstration trials were presented at the start of the clamp block.

Breaks were included throughout the experiment similar to those included in Experiment 1. They were provided for both groups at the following stages corresponding to the experimental protocol of the Test group: the middle of each clamp block (after cycles 70 and 240, 1 min), just before the start of the washout block (cycle 130, 2 min), and at 3 time points (1 min) in the washout block (cycles 167, 174, and 181).

At the end of the experiment, participants completed a short survey. To verify that the participants understood the task and followed the instructions, we asked: "Did you ever try to reach your hand to places other than the target?" To evaluate the awareness of the participants about the change in behavior, we asked: "Do you think you changed your behavior at all during the experiment?"

**Experiment 2S.** Experiment 2S was conducted to provide a replication test of the results obtained with the Test group in Experiment 2. We used an online platform created within the lab for conducting sensorimotor learning studies [32]. Participants (*N* = 48) were recruited using the crowdsourcing website Prolific (www.prolific.co). Since participants performed the experiment remotely with their personal computer, the test apparatus varied across participants. Based on self-report data, 29 participants used an optical mouse and 19 used a trackpad. Monitor sizes varied between 11 and 27 inches. Based on power calculations from the aftereffect data in Experiment 2, a minimum sample size of 9 would be required (d = 0.94). However, given the results from pilot work indicating that online data would be more variable and show a lower asymptotic level of adaptation [33], we opted to use a large sample size.

The experimental design was similar to that used for the Test group in Experiment 2 with a few minor modifications. First, the 2 Clamp blocks consisted of 80 cycles each, and the

washout block (individually determined combination of reversed-clamp and veridical feed-back trials) was extended to a total of 80 cycles. Second, we did not program in any breaks.

## Data analysis

The kinematic data recorded from the digitizing tablet were analyzed offline using custom-written MATLAB code. The primary dependent variable was the direction of hand movement (hand angle). For each trial, we identified the position of the handheld stylus when the movement amplitude was equal or larger than the radial distance to the target (8 cm). Hand angle was defined as the angle formed by a line connecting this point with the movement origin (the center of the start location) and a line connecting the target position with the movement origin. For participants who experienced a counterclockwise perturbation (rotation or clamp), the sign of the hand angle was flipped. In this manner, a positive hand angle indicates movement in the opposite direction of the perturbed feedback, the expected change due to adaptation. The mean hand angle for each movement cycle was calculated by averaging the hand angle of 4 consecutive reaches (1 reach to each of the 4 different target locations).

In all 3 experiments (Experiments 1, 2, and 2S), we observed no reliable differences on the measures of relearning between subgroups experiencing a clockwise or counterclockwise rotation and between subgroups reaching to targets positioned along the diagonal layout or oblique layout. As such, we collapsed the data across the subgroups.

For Experiments 1 and 2, all trials were included in the analysis (with 1 exception, noted below). We opted to not take any steps to exclude outliers given that participants in Experiment 1 frequently exhibit high levels of exploration after experiencing a large perturbation when instructed to focus on making the cursor hit the target. As such, we anticipated that there would be large trial-by-trial variability during the Rotation trials, at least in the first learning block, making it difficult to define, a priori, criteria for outlier removal. For consistency, we opted to also use all of the data in Experiment 2, although participants do not exhibit exploratory behavior in response to clamped feedback. We note that none of the statistical analyses were changed if repeated after exclusion of outliers (0.6% of all trials). Due to the randomization algorithm in the presentation of the Probe trials in Experiment 1, the first trial in the first (but not the second) learning block was a Probe trial. Since it was presented before any Rotation trials, the data from this trial were not included in the analysis.

We applied an outlier removal algorithm in Experiment 2S since a preliminary review of the data indicated some large deviations from the target, likely indicative of lapses of attention during the extended unsupervised testing session. We excluded trials in which the hand angle deviated from the target location by more than 100˚ and trials in which the absolute trial-to-trial change in hand angle was larger than 20˚. Based on these criteria, a total of 1.8% of all trials were excluded.

The following measures of learning were calculated: Early learning, Late learning, and Aftereffect. Separate measures were calculated for the Rotation and Probe data in each learning block of Experiment 1 and for the data in each of the 2 clamp blocks of Experiment 2 (only 1 block for the Control group). For Rotation trials in Experiment 1 and Clamp trials in Experiment 2, Early learning and Late learning were defined as the mean hand angle over cycles 3 to 7 and the last 10 cycles of each learning block, respectively (Figs 1B and 2C). Note that in Experiment 1, there is 1 cycle of Probe trials for every 4 cycles of Rotations trials. Thus, to examine learning within similar time windows for the Rotation and Probe trials, Early learning for the Probe trials was based on cycles 1 and 2, and Late learning was based on the last 2 cycles (cycles 9 and 10) in each learning block. Aftereffect was defined as the mean hand angle over the first cycle of the no-feedback aftereffect block.

Movement time was calculated as the interval between the time at which the amplitude of the movement exceeded 1 cm from the start location to the time at which the amplitude reached the radial distance of the target. Although not emphasized in the instructions, reaction time was calculated as the interval between the appearance of the target and the time that the hand position exceeded a distance of 1 cm from the start location. Total trial time was calculated as the sum of reaction time, movement time, and intertrial interval, measured as the time required to move back to the start location. For each participant, we calculated the median of each measure over all trials in a given cycle (for trial time analysis), a given learning condition/block (for reaction time analysis), or over all the trials in the experiment (for analysis of movement time).

## Statistical analysis

Two statistical approaches were used to analyze the changes in hand angle that occurred in response to the feedback perturbations. The first was based on an approach frequently adopted in the sensorimotor adaptation literature (e.g., [20,90,91]), focusing on predefined cycles to examine different phases of learning (early, late, and aftereffect). To examine changes in behavior within each of the 2 learnings blocks and between the 2 blocks, we used a 2-way repeated measures ANOVA, with 2 within-participant factors, Stage (Early and Late) and Block (Learning 1 and Learning 2). Given our focus on the between block changes (e.g., savings), we included planned comparisons, using paired-sample $t$ tests, to analyze the changes between blocks 1 and 2. We also used an independent 2-sample, between-participant $t$ test to compare the Control and Test groups of Experiment 2. Two-tailed $t$ tests were used in all of these analyses, with the statistical significance threshold set at the $p < 0.05$. For all of the $t$ tests, we report the Bayes factor $BF_{10}$, the ratio of the likelihood of the alternative hypothesis ($H_1$) over the null hypothesis ($H_0$) [92] and the Cohen's $d$ effect size [93]. For the repeated measures ANOVA, the effect size is reported using partial eta-squared ($\eta_p^2$). All of the measurements met the assumption of normality based on the Lilliefors test [94].

Although defined a priori based on previous studies, specifying a subset of the cycles as of principle interest is somewhat arbitrary and ignores much of the data. The second statistical approach was chosen to avoid these concerns. Here, we opted to use a nonparametric permutation test [24] that is widely employed in the analysis of multivariate data in which there are autocorrelations between sequential data points (e.g., as with electroencephalogram [EEG] data, see [95,96]). This "cluster analysis" approach seems well suited for the continuous and autocorrelated nature of the data obtained in studies of sensorimotor adaptation. We used this test to identify clusters of cycles in which the hand angle differed between the 2 Learning blocks and Aftereffect blocks. Two-tailed paired-sample $t$ tests were performed for each cycle within the blocks of interest. We then defined consecutive cycles in which the difference was significant ($p < 0.05$) as a "cluster," and calculated for each cluster, the sum of the t-values that were obtained for the cycles in that cluster (referred to as a t-sum statistic). A null distribution of the t-sum statistic was constructed by performing 10,000 random permutations with the data: For each permutation, the data for a given participant were randomly assigned to "block 1" or "block 2." For each permuted data set, we performed the same cluster identification procedure as was done with the actual data and calculated the t-sum statistic for each cluster. In cases where several clusters were found for a given null set permutation, we recorded the t-sum statistic of the cluster with the largest t-sum value. Thus, the generated null distribution is composed of the maximal t-sum values achieved by chance, a method that controls for the multiple comparisons involved in this analysis [24]. Each of the clusters identified in the non-permuted data was considered statistically significant only if its t-sum was larger than 95% of

the t-sums in the null distribution, corresponding to a *p*-value of 0.05. In Experiment 2, a between-subject cluster analysis was used to compare, in separate analyses, the aftereffect data from the Control group with the aftereffect data obtained from the Test group after Clamp 1 or Clamp 2 (using independent 2-sample *t* tests).

To examine within-participant changes in reaction time between the 2 learning blocks, paired-sample *t* tests were used.

### Effect size analysis of prior studies

For the review of previous work, we identified 9 papers that included conditions to provide a test of the attenuation phenomenon. Some reported multiple experiments, bringing the final total to 13 experiments. Although these experiments entail a broad range of methods, we calculated the effect size for the contrast of interest to put the results in a common context. For studies that directly compared adaptation measures between the sessions, we calculated Cohen's *d* from the reported comparisons. For the other studies, we obtained the data from the authors, performed the relevant analysis, and then computed Cohen's *d*.

Based on the design used in these studies, we classified an analysis as either early, late, aftereffect, and/or learning rate (see Fig 3) using criteria reported in the study or selected to approximate the manner in which these phases were defined in the present study. The specific criteria were the following:

- "Limited preparation time" paradigm [9]: Early learning only, comparing the first 5 trials of the perturbation phase relative to the last 5 trials of a baseline phase.

- "Aiming subtraction" [12,17,49] and "No performance error" [11] paradigms: Early and Late learning, defined as the mean hand angle over trials/cycles 3 to 7 and the last 10 trials/cycles of each learning block, respectively.

- "Implicit probing" paradigm [12,21,22]: Early and Late learning, based on the first and last 2 trials/cycles of each learning block, respectively.

- "Gradual" paradigm [50,51]: Learning rate based on model-based estimate of learning [66], calculated across the entire block. This measure reflects the proportional trial-by-trial change in hand angle with respect to the experienced error. The Aftereffect (only for [51]) defined as the mean hand angle over the first cycle of a no-feedback aftereffect block.

### Supporting information

**S1 Fig. Experiment 1: Strategy switching between Rotation and Probe trials.** Hand angle for each trial in Rotation–Probe–Rotation triplets, highlighting the switching behavior for each participant (gray lines) between the Rotation (pink dots) and Probe trials (blue dots). The horizontal dashed gray line represents full compensation for 45˚ visuomotor rotation. The data are shown for early and late stages of the 2 learning blocks. The mean values (squares) increase from early to late, and within the early phase, between the first and second block on Rotation trials, indicative of overall learning and savings, respectively. Black vertical lines indicate standard deviations. The individual data and the summary statistics presented in this figure can be found in S1 Data. The raw data can be found in https://git.io/Jtiip.
(TIF)

**S2 Fig. Experiment 2S: Replication study using an online crowdsourcing platform. (A)** Time course of participants' (*N* = 48) mean hand angle averaged over cycles, showing robust adaptation to the task-irrelevant clamped feedback (onset marked by the vertical lines) in

blocks 1 (light green) and 2 (dark green) (Early-Late Stage main effect: $F(1,47) = 90.8$, $p < 0.001$, $\eta_p^2 = 0.66$). (**B**) Overlaid hand angle functions for the 2 blocks, showing attenuation in multiple clusters throughout the learning and aftereffect blocks ($p < 0.05$). While the effect of Block was not significant ($F(1,47) = 1.09$, $p = 0.302$, $\eta_p^2 = 0.02$), there was a significant Stage × Block interaction effect ($F(1,47) = 4.99$, $p = 0.030$, $\eta_p^2 = 0.10$). (**C, D**) The interaction reflects the finding that there was a significant attenuation in the Late stage ($-1.57° ± 0.74°$, $t(47) = -2.14$, $p = 0.038$, $BF_{10} = 1.25$, $d = -0.31$), but no difference in the Early stage ($0.58° ± 0.61°$, $t(47) = 0.950$, $p = 0.347$, $BF_{10} = 0.240$, $d = 0.14$). (**E**) Similar to the results of Experiments 1 and 2, marked attenuation was observed in the Aftereffect stage ($-3.35° ± 0.81°$, $t(47) = -4.14$, $p < 0.001$, $BF_{10} = 168$, $d = -0.60$). For panels C–E, the left pair of bars show the mean across participants for each block, and the right bar shows the mean of the within-participant differences (Block 2 –Block 1). Black dots indicate individual difference scores. For all figure panels, shaded margins and black vertical lines represent SEM. The individual data and the summary statistics presented in this figure can be found in S1 Data. The raw data can be found in https://git.io/Jtiip. SEM, standard error of the mean.
(TIF)

**S1 Data. Excel spreadsheet containing, in separate sheets, the values for each individual and summary statistics for Figs 1B, 1C, 1D, 1E, 1F, 1G, 1H, 1I, 2C, 2D, 2E, 2F, 2G, 2H, 2I, 2J, 3A, 3B, 3C and 3D and S1, S2A, S2B, S2C, S2D and S2E Figs.**
(XLSX)

## Acknowledgments

We thank Maya Malaviya, Sarvenaz Pakzad, Marina Iranmanesh, and Utsav Kapoor for their assistance with data collection. We thank Assaf Breska for helpful comments on the manuscript.

## Author Contributions

**Conceptualization:** Guy Avraham, J. Ryan Morehead, Hyosub E. Kim, Richard B. Ivry.

**Data curation:** Guy Avraham.

**Formal analysis:** Guy Avraham, J. Ryan Morehead, Hyosub E. Kim, Richard B. Ivry.

**Funding acquisition:** Richard B. Ivry.

**Investigation:** Guy Avraham, J. Ryan Morehead, Hyosub E. Kim, Richard B. Ivry.

**Methodology:** Guy Avraham, J. Ryan Morehead, Hyosub E. Kim, Richard B. Ivry.

**Project administration:** Richard B. Ivry.

**Resources:** Richard B. Ivry.

**Software:** Guy Avraham.

**Supervision:** Richard B. Ivry.

**Validation:** Guy Avraham, J. Ryan Morehead.

**Visualization:** Guy Avraham, J. Ryan Morehead, Hyosub E. Kim.

**Writing – original draft:** Guy Avraham.

**Writing – review & editing:** Guy Avraham, J. Ryan Morehead, Hyosub E. Kim, Richard B. Ivry.

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
