## [Editor Report · Decision Letter 0]

31 Jul 2020

Dear Dr Avraham, 

Thank you for submitting your manuscript entitled "Re-exposure to a sensorimotor perturbation produces opposite effects on explicit and implicit learning processes" for consideration as a Short Report by PLOS Biology. Please accept my apologies for the delay in sending you the decision below to you.

Your manuscript has now been evaluated by the PLOS Biology editorial staff, as well as by an Academic Editor with relevant expertise, and I am writing to let you know that we would like to send your submission out for external peer review.

Please re-submit your manuscript within two working days, i.e. by Aug 04 2020 11:59PM.

Kind regards,

Gabriel Gasque, Ph.D.,

Senior Editor

PLOS Biology

---

## [Editor Report · Decision Letter 1]

13 Aug 2020

Dear Dr Avraham, 

Thank you for submitting your manuscript entitled "Re-exposure to a sensorimotor perturbation produces opposite effects on explicit and implicit learning processes" for consideration as a Short Report by PLOS Biology. Please accept my sincere apologies for the delay in sending this decision to you. A combination of pandemic-induced childcare duties and summer vacation absences has left us somewhat short-staffed these last two weeks, which has, unfortunately, affected our turn around times. 

In any case, your manuscript has now been evaluated by the PLOS Biology editorial staff, as well as by an Academic Editor with relevant expertise, and I am writing to let you know that we would like to send your submission out for external peer review.

Please re-submit your manuscript within two working days, i.e. by Aug 17 2020 11:59PM.

Kind regards,

Gabriel Gasque, Ph.D.,

Senior Editor

PLOS Biology

---

## [Decision Letter · Decision Letter 2]

2 Oct 2020

Dear Dr Avraham,

Thank you very much for submitting your manuscript "Re-exposure to a sensorimotor perturbation produces opposite effects on explicit and implicit learning processes" for consideration as a Short Report at PLOS Biology. Your manuscript has been evaluated by the PLOS Biology editors, by an Academic Editor with relevant expertise, and by three independent reviewers. Please accept my apologies for the delay in sending this decision to you.

In light of the reviews (below), we will not be able to accept the current version of the manuscript, but we would welcome re-submission of a much-revised version that takes into account the reviewers' comments. We cannot make any decision about publication until we have seen the revised manuscript and your response to the reviewers' comments. Your revised manuscript is also likely to be sent for further evaluation by the reviewers.

We expect to receive your revised manuscript within 3 months. 

**IMPORTANT - SUBMITTING YOUR REVISION**

Your revisions should address the specific points made by each reviewer. As you will see from the reviewers’ comments, all of them feel that the suggestion that implicit adaptation during relearning demonstrates attenuation is both interesting and thought-provoking. However, the reviewers also raise a variety of concerns, particularly with your Experiment 1, that lead them to question the strength of this conclusion. Reviewer 1 challenges the evidence for implicit adaptation, and the reviewers together ask that you provide additional support for your conclusions with additional clarifications and analyses, as well as new task designs and experimental data. Reviewer 2 also asks that you provide clarification as to whether the attenuation of implicit savings is a context-dependent/task-specific effect or is a more general phenomenon. While this reviewer indicates they’d like you to discuss this, for the broad readership of PLOS Biology, we feel that this concern would also be better assessed with additional data. We think that if you provide new and better evidence, you might be also addressing reviewer 3’s criticism.

Depending on the extent of your revision, we will decide whether we pursue your paper as a Short Report or a full Research Article upon re-submission.

Please submit the following files along with your revised manuscript:

*Re-submission Checklist*

*Published Peer Review*

*PLOS Data Policy*

*Blot and Gel Data Policy*

Sincerely,

Gabriel Gasque, Ph.D.,

Senior Editor,

ggasque@plos.org,

PLOS Biology

REVIEWS:

Reviewer #1: The authors report an interesting phenomenon in which implicit adaptation to a visuomotor rotation becomes attenuated when participants relearn the rotation after the previous adaptation is washed-out. This phenomenon stands in contrast to explicit adaptation, in which the learning becomes faster during the relearning (known as "saving"). Their experiments were carefully designed, and the data support the authors' conclusion. The manuscript is well-written. Although the mechanisms underlying the above attenuation are hypothetical ones at this stage, the findings would be provocative in the field of sensorimotor learning. Thus, this study seems to be appropriate for "Short Report" in this journal. However, there are several issues to be clarified as follows. 

Major concerns: 

1. Clamp trial as a measure of implicit adaptation

One of the critical assumptions in this study is that the performances of participants in clamp trials are measures of implicit adaptation. The authors referred to previous studies supporting this assumption. However, they need to clarify the validity of this assumption in this manuscript. 

2. Rationales for parameter settings 

It seems that the authors carefully selected parameters in the two experiments based on their long-standing experiences on the visuomotor rotations. It was not clear why they set the rotational angle to 45º in Experiment 1, or why they selected 15º as the fixed angle of the clamped feedback. 

3. An interaction effect between block and cycle

The authors stated that change in hand angle was larger early on in the second learning block, but this advantage was transient and eventually reversed in the probe trials of Experiment 1 (Lines 171 - 172). However, they need to confirm a significant interaction between block (learning or relearning) and cycle before this statement. 

Minor points:

Line 157: "a point that that would be farther …" This sentence was not understandable. 

Line 539 - 540: "… the second learning block, These probes …" Should the comma be replaced by a period?

Reviewer #2: The current study investigated the effect of explicit and implicit components on savings, the phenomenon consistently reported in sensorimotor re-learning. With a series of elaborated visuomotor adaptation experiments to rotated cursor feedback, the authors have found that the implicit adaptation has the attenuated effect on the savings, which contradicts the previous reports. 

By using a novel experimental method, the authors provide essential findings related to mechanisms about savings of visuomotor adaptation. The manuscript is well written, and the methodological approach and results are well explained. I have just two substantive points to address in revision: 

Major comments: 

1. The authors found that the implicit component is attenuated upon re-learning. Their finding is somehow surprising because this is incongruent with the recent papers which reported the implicit adaptation has a positive effect on saving. The authors have already acknowledged this fact by citing these studies and mentioned this discrepancy in the Discussion section. I wonder if the observed finding of attenuated implicit saving is context-dependent and specific to their current experimental setups, being not generally applicable to sensorimotor learning. The authors should discuss their current findings more in detail by explaining the specific mechanisms observed in their study. 

2. One possible concern throughout the experiments is whether the participants accurately followed the given instructions. For example, in experiment 1, the authors used the two target colors (red or blue) to instruct the participants to explicitly use the different strategies. Although the authors emphasized the instructions by displaying the message during the task (experiment 1) or performing the demonstration of the clamped feedback trials (experiment 2), I wonder if these instructions are really followed by the participants. In addition, the instruction, target "will act differently", provides ambiguous information, and how the participants understood this instruction would affect their strategy. These concerns were not entirely addressed in this manuscript. 

Minor comments:

3. Please describe how the sample size was determined for each experiment.

4. I wonder if there is any difference between clockwise versus counterclockwise perturbations. 

5. It is not clear how the authors determined the number of trials for the extended washout block. 

6. The last part of the Result section (page14, line 250-; "This attenuation of implicit learning, …") should better be moved to the Discussion part. 

Reviewer #3: 

This well-written paper shows the results of two experiments and describes the results from three previous studies that point to the attenuation in the implicit component of visuomotor adaptation upon relearning. The study is new and provoking because the existence of savings in visuomotor adaptation is commonly accepted, at least in abrupt perturbation conditions. The authors thus propose that only the explicit component exhibits savings. The results of experiments 2 are rather convincing, although the effect sizes (not provided) appear small. I do have however important reservations with the strength of the effect, notably in experiment 1. 

Major

Both the design and the results of experiment 1 are relatively weak. About the design, I agree with the authors that the "failure to completely dispense with the aiming strategy would contaminate the probe trials" (line 180). This design thus creates a problem in the pure assessment of the implicit component. About the results, I do not agree that they "provide clear evidence of savings". The reduced savings are indirectly inferred by the small between-group differences in the no-feedback block (figure 1I suggests a large variance, thereby a small effect size). In fact, the results show a small but significant increase, not decrease, in savings at the beginning of the relearning block. 

Along the same lines, the effect sizes for the "prior evidence for attenuation upon relearning for implicit visuomotor adaptation" appear small. The main worry here is cherry-picking those adaptation studies that have shown an attenuation effect and ignoring those that have shown no effect or a savings effect. A well-designed comprehensive meta-analysis is needed here to guard against a potentially premature conclusion. Such meta-analysis would include studies in which implicit savings have been shown (e.g., Coltman et al., 2019; Yin and Wei, 2020), studies in which no savings have been shown (e.g. Haith et al., 2015; Leow et al., 2020), and other savings studies that are not included (e.g., Klassen, et al. 2005; Orban de Xivry and Lefèvre 2015; Oh and Schweighofer 2019).

Finally, the implicit component of visuo-motor adaptation is thought to be a cerebellar-dependent process. Because the results presented are opposite to previous results showing savings in other cerebellar-dependent processes (such as eye-blink conditioning and saccadic adaptation, as cited), additional solid evidence from well-designed experiments that specifically aim at studying savings in the implicit component, such as experiment 2, are needed. 

Minor

The two sub-groups of experiment 1 are not justified. Indeed, the rotation and target positions of the first sub-group are not ideal, since the "correct answers" are shooting to the cardinal directions. Too much strategy is involved here. Is there a difference in results between the sub-groups?

Only the results of statistical tests are reported. We need to see means, standard deviations, and effect sizes. The presumably small effect sizes overall in both experiments are troubling from a reproducibility point of view.

The main statistical test used is the uncorrected t-test. Given the large number of tests performed there is a large probability of chance findings. Consider using a correction such as the FDR. In addition, it is unclear whether the normality assumptions have been verified. 

I like in principle the permutation approach, but it also seems to pick up noise. See for instance the small cluster of significant difference around 25 trials in Figure 1D. Showing the p values instead of the p<0.05 threshold, as well as the effect size, may help here. 

Line 73: Neither I, nor the authors in line 180, agree with this statement

---

## [Decision Letter · Decision Letter 3]

5 Feb 2021

Dear Dr Avraham,

Thank you for submitting your revised Short Report entitled "Re-exposure to a sensorimotor perturbation produces opposite effects on explicit and implicit learning processes" for publication in PLOS Biology. I have now obtained advice from the original reviewers and have discussed their comments with the Academic Editor. You will note that reviewers 1 and 2, Hiroshi Imamizu and Kenji Ogawa, respectively, have signed their comments.

Based on the reviews, we will probably accept this manuscript for publication, assuming that you will modify the manuscript and the associated metadata to address the remaining editorial request listed below my signature.

We expect to receive your revised manuscript within two weeks. 

*Published Peer Review History*

*Early Version*

Sincerely,

Gabriel Gasque, Ph.D.,

Senior Editor,

ggasque@plos.org,

PLOS Biology

ETHICS STATEMENT:

-- Please include in your manuscript the ID number of your protocol approved by the Institutional Review Board at the University of California, Berkeley.

-- Please indicate in your manuscript if you approved protocols adhered to the principles expressed in the Declaration of Helsinki or to any other national or international ethical guidelines.

DATA POLICY:

I note that you have provided your data in the GitHub repository: https://github.com/guyavr/ReexposureSensorimotorPerturbationProducesOppositeEffectsExplicitImplicit”

Please update the README file to include detailed information on how your data were analyzed to generate the plots shown in figures 1B-I, 2CJ, 3A-D, S1, and S2A-E.

Alternatively (or in addition) you can provide a spread sheet with all individual quantitative observations used to generate the plots listed above. For an example see here: http://www.plosbiology.org/article/info%3Adoi%2F10.1371%2Fjournal.pbio.1001908#s5

The numerical data provided should include all replicates AND the way in which the plotted mean and errors were derived (it should not present only the mean/average values).

Please also ensure that each figure legend in your manuscript includes information on where the underlying data can be found and that your supplemental data file/s has/have a legend.

Reviewer remarks:

Reviewer #1, Hiroshi Imamizu: The authors fully addressed my previous concerns. 

Reviewer #2, Kenji Ogawa: I found the resubmitted manuscript to be revised properly regarding all the comments that I had made on the original version, thus recommending the acceptance for publication. 

Especially, I'm happy to know that their results were replicated by their novel online platform experiment, which further validates their findings. 

Reviewer #3: The authors have adequately addressed all my comments. In particular, the new replication experiment and the (now) in-depth literature review (with effects sizes) largely strengthen the paper.

---

## [Editor Report · Decision Letter 4]

15 Feb 2021

Dear Dr Avraham,

On behalf of my colleagues and the Academic Editor, Alexander Gail, I am pleased to say that we can in principle offer to publish your Short Report "Re-exposure to a sensorimotor perturbation produces opposite effects on explicit and implicit learning processes" in PLOS Biology, provided you address any remaining formatting and reporting issues. These will be detailed in an email that will follow this letter and that you will usually receive within 2-3 business days, during which time no action is required from you. Please note that we will not be able to formally accept your manuscript and schedule it for publication until you have made the required changes.

PRESS

Thank you again for supporting Open Access publishing. We look forward to publishing your paper in PLOS Biology. 

Sincerely, 

Gabriel Gasque, Ph.D. 

Senior Editor 

PLOS Biology